5

# Relating historical vegetation cover to aridity index patterns in the greater desert region of northern China: Implications to planned and existing restoration projects

Yanying Shao<sup>1</sup>, Yuqing Zhang<sup>1,2</sup>, Xiuqin Wu<sup>1,3</sup>\*, Charles P.-A. Bourque<sup>1,4</sup>, Jutao Zhang<sup>1</sup>, Shugao Qin<sup>1,2</sup>, and Bin Wu<sup>1,2</sup>

<sup>1</sup>Yanchi Research Station, School of Soil and Water Conservation, Beijing Forestry University, Beijing 100083, PR China <sup>2</sup>Key Laboratory of State Forestry Administration on Soil and Water Conservation, Beijing Forestry University, Beijing 100083, PR China

<sup>3</sup>Engineering Research Center of Forestry Ecological Engineering, Ministry of Education, Beijing Forestry University, Beijing
 100083, PR China

<sup>4</sup>Faculty of Forestry and Environmental Management, University of New Brunswick, New Brunswick E3B 5AS, Canada *Correspondence to*: Xiuqin Wu (wuxq@bjfu.edu.cn)

Abstract. Desert regions of northern China have always been the most severely affected by climate change, especially in terms of their ecological integrity and social sustainable development. Assessments of dryness in both space and time are central to

- 15 the development of adaptation strategies to climate change. Earlier studies have identified long-term patterns of dryness in northern China, but these studies have usually been of limited value to land-management planning as they ignore local-toregional-scale climate features. To identify potential cause-and-effect relationship between aridity and vegetation cover, changes in aridity index (AI) and vegetation cover were tracked with the assistance of a chronological series of surfaces based on the mapping of AI and normalized difference vegetation index (NDVI) and convergent cross mapping. By tracking regional-
- scale variation in precipitation, air temperature, AI from 1961-2013 (53 years), and vegetation cover dynamics from 1982-2013 (32 years), we show that precipitation increased in approximately 70% of the greater desert region, including in the Ulanbuh, Tengger, Badain Jaran, Qaidam, Kumtag, Gurbantunggut, and Taklimakan Deserts. This increase was statistically strongest for the Gurbantunggut (p 

#### **1** Introduction

In recent decades, countless studies have demonstrated the indisputable fact of climate change (IPCC, 2013) and its impacts on water resources (Raghavan et al., 2012; Deng et al., 2015; Pumo et al., 2016), ecological environment (Chen et al., 2013; Yang et al., 2015), human settlement (Dumenu and Obeng, 2016), and human health (Wu et al., 2016; Moore et al., 2008;

5 Abaya et al., 2011). Continued global climate change in desert regions of the world may have a profound impact by accelerating hydrological processes and increasing the unpredictability of related hydrometeorological variables (Gan, 2000; Ma et al., 2004; Jentsch and Beierkuhnlein, 2008), leading to a re-shaping of desert-dryness patterns.

Desert regions of northern (N) China cover nearly one-fifth of the country's land base. Drylands have always been the most severely affected by changes in climate (Fullen and Mitchell, 1994; Liu and Diamond, 2005; Wang et al., 2013). Levels of

10 dryness are closely related to desertification and their assessment are fundamental to the development of adaptation strategies. Therefore, an understanding of drying and wetting conditions and their relationship to climate change is vital for the sustainable development of fragile desert environments.

Prior to the 1970's, greater amounts of precipitation fell in the Inner Mongolia region and Xinjiang region; however, since the 1980's, precipitation has decreased by 15-40 and 0-10 mm yr<sup>-1</sup> in the Inner Mongolian Plateau and southern Xinjiang

- <sup>15</sup> province, respectively (Xu et al., 2010). Li et al. (2013) reported that precipitation had increased 10.2, 6.3, and 0.9 mm 10-yr<sup>-1</sup> in the last 50 years in the mountains, oases, and deserts of northwest (NW) China. In general, precipitation in NW China increased 0.61 mm yr<sup>-1</sup> (p 

air temperature and reveal pattern changes in aridity, and (2) track spatiotemporal changes in vegetation dynamics with respect to plant growth and plant-cover expansion and contraction in corroborating our assessments of climatic change.

#### 2. Materials and methods

#### 2.1. Study regions

- Desert regions of N China (75°-125 E longitude, 35°-50 N latitude) cover approximately 172.1 × 10<sup>4</sup> km<sup>2</sup>, accounting for about 17.9% of the total land area of China (State Forestry Administration, P.R. China, 2015a). The land area forms a discontinuous arc-shaped desert belt from the western Tarim Basin to western Songnen Plain. It traverses the NW-, N-, and NE-regions of China with a length of 4,500 km from east to west and a width of 600 km from south to north, including most of Xinjiang, Inner Mongolia, Ningxia, N Gansu, Hebei, Heilongjiang, Jilin, Liaoning, and a part of Qinghai and Shaanxi
- provinces (Zhu and Liu, 1981; Li et al., 2013; Li et al., 2014). These areas are characterized by different climate types, from hyperarid (AI < 0.05), arid ( $0.05 \le AI < 0.2$ ), semiarid ( $0.2 \le AI < 0.5$ ), to dry subhumid climate types ( $0.5 \le AI < 0.65$ ; Hulme, 1996; Middleton and Thonas, 1997). The deserts span two vegetation zones; steppes in the east and temperate deserts in the west of the Helan Mountain Range (106 °E longitude). Harsh natural conditions (e.g., drought, wind erosion, sparse vegetation, and infertile soils), combined with anthropogenic undertakings (e.g., overgrazing, excess land reclamation, irrational
- exploitation of water resources, and so on), has resulted in serious land degradation and desertification in many parts of these regions. To improve ecological conditions, the Chinese government's "Grain for Green" program (1999-present) was responsible in re-planting vast areas of N China. By the end of 2014, 15.93 × 10<sup>4</sup> km<sup>2</sup> of N China had been converted into forests and grassland (State Forestry Administration, P.R. China, 2015b).

#### 2.2. Data source and processing

- By means of desert (Zhu et al., 1980) and sand-covered desert classification maps of China, at a 1:100,000 scale (http://westdc.westgis.ac.cn), we were able to delineate boundaries of twelve deserts, namely the Hulun Buir, Horqin, Otindag, Hobq, Mu Us, Ulanbuh, Tengger, Badain Jaran, Qaidam, Kumtag, Gurbantunggut, and Taklimakan Deserts (Fig. 1). Timeseries of mean, minimum, and maximum air temperature, air pressure, wind speed, relative humidity, sunshine hours, and precipitation from January 01, 1961 to December 31, 2013, were collected from 113 meteorological stations (Fig. 1)
- through China's Meteorological Data Sharing Service System (http://www.escience.gov.cn/metdata/page/index.html). Data with respect to vegetation was based on 15-day resolution timeseries of Global Inventory Modeling and Mapping Studies (GIMMS) NDVI, with an original spatial resolution of 8 km, http://ecocast.arc.nasa.gov/data/pub/gimms/3g.v0/. Spatial interpolation of meteorological data is based in part on an 8-km re-sampling of an existing 1-km resolution digital elevation model (DEM) developed by the Heihe Project Data Management Centre (http://heihedata.org/).
- Incomplete timeseries were gap filled with linear regression with complete timeseries from adjacent meteorological stations serving as independent variable. To assess the spatial variation in annual trends in precipitation, air temperature, and AI, spatial interpolation of meteorological data was performed at a spatial resolution of 8 km using the thin-plate smoothing spline method

of Hutchinson (2004; http://cres.anu.edu.au/). We generated annual composites of NDVI from the 15-day images with the maximum value composite method.

#### 2.3. Aridity Index

5

15

Aridity index (AI) is used to analyse the spatiotemporal variation of dryness in desert regions of N China. It was calculated from:

$$AI = \frac{P}{PET},$$
(1)

where *P* is the annual precipitation (mm) and *PET* is the annual potential evapotranspiration (mm) based on the summation of daily reference evapotranspiration (i.e.,  $PET_o$ , in mm d<sup>-1</sup>) calculated with the Penman-Monteith equation (Allen et al., 1998), i.e.,

10 
$$PET_o = \frac{0.408\Delta(R_n - G) + \gamma \frac{900}{T + 273} u_2(e_s - e_a)}{\Delta + \gamma (1 + 0.34 u_2)},$$
 (2)

where Rn and G are the net radiation and soil heat flux (in MJ m<sup>-2</sup> d<sup>-1</sup>),  $\gamma$  is the psychometric constant (kPa °C<sup>-1</sup>),  $e_s$  is the saturation vapor pressure (kPa),  $e_a$  is the actual vapor pressure (kPa),  $\Delta$  is the slope of the saturation vapor pressure-vs.-air temperature curve (kPa °C<sup>-1</sup>), T is the average daily mean air temperature (°C), and  $U_2$  is the mean daily wind speed at a 2-m height (m s<sup>-1</sup>). Computation of variables in Eq. (2) follow the approaches used by the Food and Agriculture Organization of the United Nations (FAO). The main radiation component of Eq. (2) is calculated from:

$$R_n = R_{ns} - R_{nl} , (3)$$

$$R_{ns} = (1 - \alpha)R_s \,, \tag{4}$$

$$R_s = \left(a + b\frac{n}{N}\right)R_a\tag{5}$$

$$R_{nl} = \sigma(\frac{T_{xk}^4 + T_{nk}^4}{2}) \left( 0.34 - 0.14\sqrt{e_a} \right) \left( 1.35 \frac{R_s}{R_{s0}} - 0.35 \right),\tag{6}$$

- 20 where  $R_a$  is the extraterrestrial radiation,  $R_s$  is the incident solar radiation,  $R_{so}$  is the clear-sky solar radiation,  $R_{ns}$  and  $R_{nl}$  are the net shortwave and longwave radiation (all, in MJ m<sup>-2</sup> d<sup>-1</sup>),  $\alpha$  is the albedo ( $\alpha = 0.23$  for desert environments, Warner, 2004), *n* and *N* are the actual and maximum possible sunshine duration (h),  $\sigma$  is the Stefan-Boltzmann constant (4.903×10<sup>-9</sup> MJ K<sup>-4</sup> m<sup>-2</sup> d<sup>-1</sup>),  $T_{xk}$  and  $T_{nk}$  are the maximum and minimum absolute temperature during any 24-h period (K), and *a* and *b* are empirical coefficients (i.e., *a* = 0.161 and *b* = 0.614, after Wang et al., 2014). Variables  $R_a$ ,  $R_{so}$ , and *N* were calculated from
- solar constant, latitude, elevation, and the number of days in the year following procedures outlined in the FAO56 report.

### 2.4. Mann-Kendall test

To detect changes in precipitation, air temperature, and AI in the twelve deserts, the Mann Kendall (M-K) non-parametric test was used. For a timeseries of "n" observations (i.e., $x_1, x_2, ..., x_n$ ), the M-K test is based on the S-statistic, which is computed from

$$S = \sum_{i=1}^{n-1} \sum_{j=i+1}^{n} sgn(x_j - x_i),$$
 (7)

where

$$\operatorname{sgn}(x_{j} - x_{i}) = \begin{cases} 1 & x_{j} > x_{i} \\ 0 & x_{j} = x_{i} \\ -1 & x_{j} < x_{i} \end{cases}$$
(8)

When the sample size  $(n) \ge 8$ , the variance of the S-statistics is calculated from

$$Var(S) = \frac{n(n-1)(2n+5) - \sum_{k=1}^{m} t_k(t_k-1)(2t_k+5)}{18}$$
(9)

(Mann, 1945; Kendall, 1975), where "m" is the number of tied groups,  $t_k$  denotes the number of ties for  $x_j > x_i$ . A tied group is a set of sample data having the same value. The standardized M-K test statistic is obtained from

$$Z = \begin{cases} \frac{S-1}{\sqrt{Var(S)}} & S > 0\\ 0 & S = 0.\\ \frac{S+1}{\sqrt{Var(S)}} & S < 0 \end{cases}$$
(10)

If  $|Z|>Z_{1-p/2}$ , a significant trend exists in the timeseries. If |Z| > 1.64, it indicates a statistically significant trend at a significance level of p = 0.1; whereas |Z| > 1.96 and > 2.576 denote statistically significant trends at p = 0.05 and 0.01,

respectively. A positive value of Z indicates an 'upward trend', whereas a negative value indicates a 'downward trend'; zero corresponds with 'no statistically detectable trend'. To estimate the actual rate-of-change (i.e., trend) in the data we employ localized linear regression at individual image pixels. By definition, the slope of the regression line is equivalent to the rate-of-change in the timeseries data.

#### 2.5. Convergent cross mapping

- Convergent cross mapping is a model-free method that detects cause-and-effect relationships and direction of cause-and-effect in dynamic systems (Sugihara et al., 2012). As a general rule, timeseries are considered causally related if both timeseries originate from the same system. Convergent cross mapping tests for cause-and-effect by determining the extent historical records in one timeseries can reliably determine the state in a second timeseries. The method provides consistent description of cause-and-effect even in the presence of system feedback and confoundedness (Sugihara et al., 2012). Moreover, convergent
- cross mapping involves convergence, a distinct feature of the method that distinguishes cause-and-effect from conventional

correlation. Ordinarily, non-causal associations are illustrated as horizontal, non-convergent curves of predictive skill generated from calculations of Pearson's correlation coefficient between predictions and actual observations as timeseries record lengths are extended. Cause-and-effect relationship between variables is inferred when convergence is present and Pearson's correlation coefficient at the point of convergence is > 0.0.

#### 5 3. Results

10

#### 3.1. Precipitation

Figure 2 provides the annual precipitation from 1961-2013 for the twelve deserts. The results indicate annual fluctuations in precipitation, with an obvious drop in precipitation from east to west. According to the M-K test, precipitation increased for seven of the twelve deserts, namely for the Ulanbuh, Tengger, Badain Jaran, Qaidam, Kumtag, Gurbantunggut, and Taklimakan Deserts. A statistically strong increasing trend was observed for the Gurbantunggut (p < 0.01) and the Taklimakan Deserts (p < 0.1). In contrast, five of the deserts (i.e., Hulun Buir, Horqin, Otindag, Hobq, and Mu Us) demonstrated declining trends in precipitation. However, these trends were not statistically significant (Table 1).

In addition, slopes in precipitation exhibited increasing trends in seven of the deserts, in order from lowest to highest, Kumtag, Tengger, Badain Jaran, Qaidam, Ulanbuh, Taklimakan, and Gurbantunggut Deserts, with a rate-of-change of 0.05-

1.21 mm yr<sup>-1</sup>, and decreasing trends in Hobq, Otindag, Mu Us, Hulun Buir, and Horqin Deserts, with a rate-of-change of
 0.17, -0.33, -0.35, -0.54, and -0.88 mm yr<sup>-1</sup>, respectively (Table 1).

Results of linear regression on annual precipitation at the pixel-level are summarized in Fig. 3. The results show that during the observation period (1961-2013), an increasing trend in precipitation occurred within about 70% of N China, mainly in the western half of the greater desert region (Fig. 1). The trends are statistically significant and strongest in the northern-half of

20 the western desert region (p < 0.05); whereas, a decreasing, though not statistically significant, trend (i.e., p > 0.05) was observed to have occurred in the eastern part of the study area, affecting about 30% of the greater desert region.

#### 3.2. Mean air temperature

Timeseries of annual mean air temperature (1961-2013) for the twelve deserts are shown in Fig. 2. The M-K tests show that all twelve deserts exhibited a statistically significant upward trend in air temperature (p < 0.01; Table 1). Similar trends were

25 detected by linear regression (Table 1). Highest rate of air temperature change was observed to have occurred in the Hobq Desert at  $0.082^{\circ}$ C yr<sup>-1</sup>, amounting to about 6.3 times greater than the lowest rate determined for the Ulanbuh Desert (i.e.,  $0.013^{\circ}$ C yr<sup>-1</sup>). Figure 4 shows the changes in annual mean air temperature over the various deserts. Unsurprisingly, 97% of the greater desert region was associated with a distinct rising trend in annual temperature (p < 0.05) with the majority of the area observing increases of  $0.02-0.04^{\circ}$ C yr<sup>-1</sup>.

#### 3.3. Aridity

Timeseries of annual AI show annual fluctuations, with a strong drying trend from east to west (Fig. 2). According to M-K tests, the Ulanbuh, Badain Jaran, Qaidam, Kumtag, Gurbantunggut, and Taklimakan Deserts exhibited wetting trends, which were especially strong for the Gurbantunggut (p < 0.01), Taklimakan (p < 0.05), and Qaidam Deserts (p < 0.05). Drying trends

observed for the Hulun Buir, Horqin, Otindag, Hobq, Mu Us, and Tengger Deserts were not statistically significant at p = 0.05 (Table 1).

Results from linear regression supported our earlier trend analysis with a rate-of-change of 0.0006-0.013 per decade. The highest positive rate-of-change in AI was for the Gurbantunggut Desert (0.0013) and the lowest for the Kumtag Desert (0.00006). A declining (i.e., drying) trend was detected for the Hulun Buir, Horqin, Otindag, Hobq, Mu Us, and Tengger

Deserts with a rate-of-change of -0.00014 to -0.008 per decade.

Overall, the results indicated that AI in more than half of the deserts (63.6%; Fig. 1) exhibited a wetting trend (i.e., increasing AI); this trend, for the most part, was statistically significant. About 36.4% of the affected area exhibited a drying trend (decreasing AI), though not statistically significant (Fig. 5). This suggests that most of the desert belt of N China is experiencing some level of wetting. As shown in Fig. 5, the wetting zones were mainly distributed in the western desert region (p < 0.05).

The higher rates-of-change appear in the NW of the belt, with a rate-of-change > 0.005 per decade.

#### 3.4 Vegetation dynamics

Spatial distribution of annual mean NDVI from 1982-2013 largely shows a gradual reduction in vegetation vigor and cover from east to west, with the higher values of NDVI occurring in the oases within the western part of the greater desert region (Fig. 6). Areas of annual mean NDVI 

#### 4. Discussion

5

According to global trends, precipitation has generally increased over the Northern Hemisphere at middle to high latitudes, with wet areas becoming wetter and dry areas becoming drier (New et al., 2001; Dore, 2005). Our results suggest an opposite trend, where dry areas have become wetter and wet areas have become drier, although not statistically significant in all cases. Clearly, these findings emphasize the complexity of short- to long-term spatiotemporal patterns in meteorological/climatological processes in desert regions. According to the IPCC (2013), climatic variability combined with human-induced emission of greenhouse gases has resulted in an increase of near-surface air temperatures of 0.72°C during the

1951-2012 period, amounting to a 0.01°C yr<sup>-1</sup> increase. Our study shows that the effect of global climate change has resulted in far greater increases in the deserts of N China, averaging to about 0.025°C yr<sup>-1</sup> from 1961-2013. This increase was about 2.5
times greater than the global rate. In terms of aridity, our results are consistent with findings previously reported for NW China

and semiarid regions of Inner Mongolia (e.g., Huo et al., 2013; Zhou et al., 2015). Historical trends in NW China are in contrast to the 'dry gets drier and wet gets wetter' association that we currently see globally (Melillo et al., 1993; Herrmann et al., 2005; Whitford, 2002; Lioubimtseva, 2004; Donohue et al., 2009).

Precipitation is an important driving factor for vegetation growth and development in drylands (Cleland et al., 2007; Zhang

et al., 2006; Xin et al., 2008; Fig. 8). In general, increased precipitation and aridity index is followed by improvements in vegetation cover (Zhang et al., 2013). In our study, historical trends in NDVI exhibited obvious spatial variations with a decreasing trend from east to west, analogous to the patterns we observe in precipitation (Fig. 6). Elevated air temperatures may cause the melting of high-elevation glaciers (a major source of water in the inland arid areas of NW China; Wang et al., 2008; Sorg et al., 2012; Hua and Wang, 2014; Matin and Bourque, 2015) to accelerate (Zhang et al., 2016), potentially

increasing the amount of water available for oasis agriculture and vegetation establishment.

Matin and Bourque (2015) show through their work that vegetation cover dynamics and surface evaporation in the lowland oases of two large endorheic (hydrologically-closed) watersheds of central Gansu (expressed in terms of satellite-based estimates of enhanced vegetation index) was causally related to the production of precipitation in the Qilian Mountains (south of the oases) and precipitation in the mountains was causally related to the seasonal phenology and coverage of oases vegetation,

- with the former relation providing the stronger feedback control. Convergent cross mapping of within-oasis production of precipitation through localized convective processes is shown to have no causal relation with NDVI in the oases (i.e., p > 0.05and convergence is absent). This assessment is consistent with the view that in-mountain production of precipitation through orographic-lifting of evaporated water from the oases and surface and shallow subsurface return flow from the mountains is central to sustaining healthy growing vegetation in the oases (Bourque and Matin, 2012; Matin and Bourque, 2015). Added
- water resources to the oases in the form of glacial meltwater should help sustain the oases for some time into the future, providing that the vegetation cover in the oases (i.e., transpiration pump) is not radically altered by urbanization and other forms of land conversion (Bourque and Hassan, 2009). For semiarid regions of NE China undergoing drying, rising air

temperatures may accelerate surface evaporation (Hao et al., 2012; Xin et al., 2008) and cause vegetation vigor and cover to decline, potentially promoting desertification.

Although impacts of precipitation and air temperature on ecosystems are important, ecological processes of ecosystems are largely influenced by their integrated effects (aridity vs. vegetation, Fig. 8). Spatiotemporal patterns of aridity determine the

- 5 spatial structures and evolution of plant communities (Fernandez-Illescas and Rodriguez-Iturbe, 2004; Oguntunde et al., 2006; McVicar et al., 2007; Jiapaer et al., 2015; Fig. 8) and the process of desertification in arid-to-semiarid regions (Wang, et al., 2009). As illustrated in Fig.8, annual changes in NDVI are shown to be controlled to a large measure by annual changes in aridity (p 

of this study (8 km), aridity controls the majority of change in vegetation cover (NDVI), with no statistical support of feedback between these two variables. Trend analysis of precipitation, aridity, and NDVI suggest that climatic conditions in the western desert region in the past 53 years has undergone a level of wetting, leading to small improvements in vegetation cover (NDVI). In contrast, semiarid areas of the eastern desert region have experienced declining ecological conditions over the same time period.

Our results are critical for adaptation planning and desertification management which can provide guidance to the ecological restoration of deserts in N China. However, promoting sustainable management of re-vegetation projects during the next cycle of restoration planning remains a major challenge. We recommend that more information, such as tolerance of vegetation cover in an environment of increasing demand for water by residential, agricultural, and industrial sectors, be obtained to

10 ensure socio-economic and ecological sustainability of dryland systems. Although, vegetation cover is explicitly not shown to affect aridity by feedback mechanisms at the current resolution, we suggest that further work be done to uncover these relationships as over-exploitation of the land vegetation may lead to further decline, even in instances where precipitation may be trending slightly upwards.

#### 6. Data availability

Timeseries of mean, minimum, and maximum air temperature, air pressure, wind speed, relative humidity, sunshine hours, and precipitation from January 01, 1961 to December 31, 2013, were obtained from China's Meteorological Data Sharing Service System (http://www.escience.gov.cn/metdata/page/index.html); Global Inventory Modeling and Mapping Studies (GIMMS) NDVI were accessed from http://ecocast.arc.nasa.gov/data/pub/gimms/3g.v0/; while the digital elevation model was obtained from the Heihe Project Data Management Center (http://heihedata.org/).

#### 20 Author contributions

YQZ and YYS designed the study of the paper; YYS, YQZ, XQW, and CPAB contributed to the ideas, interpretation of the data, and manuscript writing; CPAB employed CCM in a cause-and-effect analysis of precipitation, AI, and NDVI; YYS contributed to data processing and analysis; all authors contributed to the discussion and commented on the manuscript throughout its development.

#### 25 Conflicts of Interest

The authors declare no conflict of interest.

#### Acknowledgments

The authors gratefully acknowledge the financial support provided by National Key Basic Research Program of China (No. 2013CB429901) and National Key Research and Development Program of China (No. 2016YFC0500905).

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
