# Peer review of "Relating historical vegetation cover to aridity index patterns in the greater desert region of northern China: Implications to planned and existing restoration projects"

_Biogeosciences, 2016_

## Referee Comment (RC1) · Anonymous Referee #1 · 24 Oct 2016

Comment The paper addressed relevant scientific questions within the scope of BG. Based on long-term meteorological records, this research work analyzed the impact of climate change on the trends in aridity (aridity index), and vegetation cover (satellite-based NDVI) in the greater desert belt of N China. Trend analysis indicated that the overall environmental conditions were improved in the western desert and declined in the eastern China. The cause-and-effect relationship analysis indicated that aridity controls the majority of change in vegetation cover at the 8-km spatial resolution of this study. This information is useful for re-vegetation projects in this area. This finding is novel.

[Figure]

Suggestion From the aridity index trend (Fig. 5), the west was getting wetter and the east was getting drier. From NDVI trend (Fig 7), the vegetation had the improving trend in the west, but there are some scattered red regions, which denote the vegetation was declined in these places. The discrepancy between climate becoming wetter and vegetation decline in these local areas could be caused by revegetation programs, which over exploited ground water and cause natural vegetation nearby declined. If this is true, it will support the recommendation of the paper that more information, such as tolerance of vegetation cover in an environment of increasing demand for water by residential, agricultural, and industrial sectors, be obtained to ensure socio-economic and ecological sustainability of dryland systems. I would suggest the authors discuss it in the manuscript.

---

## Referee Comment (RC2) · Anonymous Referee #2 · 27 Oct 2016

The desert ecosystems are vulnerable to climate change, particularly its effect on vegetation development. Previous study on climate change in desert regions of north China was mostly focused on large-scale modelling of arid climate features. This manuscript provides a local to regional-scale analysis of climate changes in the greater desert belt of north China, based on long-term meteorological records, aridity index (AI) and satellite-based NDVI calculation. The obtained results are quite interesting, particularly the change in precipitation which shows "dry areas becoming wetter and wet areas becoming drier" in the north China deserts, an opposite trend compared to global observations. The research plan in this manuscript is sound and overall presentation is

well structured. However, some weaknesses may need to be clarified/ improved. 1. In general, the writing quality in the sections of "Introduction" and "Results" is obviously low, compared with Abstract and Discussion. It would be worth improving the writings accordingly. My specific suggestions are to:

1) Re-write the paragraph at Page 2 Line 13 – 20. Maybe start with a key sentence stating general research findings and pay an attention on linkage of context. 2) Revise the sentence at Page 6 Line 19 – 21, such as: "The increasing trend is statistically significant and strongest in the northern-half of the western desert region (p < 0.05). Comparably, a decreasing trend, though not statistically significant (p > 0.05), was observed in the eastern part of the study area, affecting about 30% of the greater desert region." 3) Regulate the inconsistent order of references cited in the text (see Page 2 Line 2-5).

2. References of No. 3 and 5 (Page 11 Line 10 and Line 15) seem to be unpublished reports. Unsure whether or not it is necessary to list them in this paper. 3. Page 3 Line 15 "has resulted" should change to "have resulted".

---

## Referee Comment (RC3) · Anonymous Referee #1 · 3 Nov 2016

After some reflection and as a follow-up to my earlier comments regarding the manuscript, "Relating historical vegetation cover to aridity index patterns in the greater desert region of northern China: Implications to planned and existing restoration projects", I pose several questions for the authors' to consider in their revision of the manuscript. In general, I still feel the manuscript has plenty to contribute to the scientific literature, but I believe some of the key parts of the paper could be improved. I believe that some of these questions/comments may help to elevate the quality of the manuscript. Some general questions: 1. What are the merits of the research? Is the merging of AI calculations with space-borne evidence of vegetation change novel

here? What about the usage of convergent cross mapping? Are the results of the analysis supported by prior studies of this kind? The authors should highlight them in the discussion. 2. In terms of convergent cross mapping and the lack of feedback between AI and NDVI, does this support the current state of knowledge about the relationship between these two variables? The authors should try to expand/elaborate on the extent that this may or may not be true. The proposed reason of spatial resolution coarseness may be right, but is this understanding supported by the scientific literature? Corroborative evidence in the existing literature would help firm this explanation as a possibility; so, more insight on this issue is needed in the discussion part. 3. The statements regarding existing and planned re-vegetation projects. Is there any evidence within the desert belt at the field level that supports some of the conclusions reached in the manuscript, i.e., wetting and vegetation improvement in the west and drying and vegetation decline in the east. Field-based evidence showing some of these trends would help solidify the paper. 4. According to Regional Climate Model or Global Climate Model projections for N China, are the expectations for western (wetting) and eastern (drying) N China borne out in their future climate projections for the region. The reason why I ask is because the study as you know is based on historical trends; is there any expectation that these trends will continue into the future? An evaluation of modelled trends for the area would help firm up the importance of past trends in describing the potential risk of future climate changes to the persistence of existing and planned re-vegetation projects. If there are no expectations that these trends will continue, what is the value of the current research regarding existing and future projects. In the manuscript, the authors state that the trends will continue. Is this supported in the scientific literature and/or climate change projections for the region?

Specific questions: 1. Page 5, line 10. How are the numbers of "m" and "tk" calculated? The authors should give more explanation. 2. Page 5, line 19. The method of convergent cross mapping is well addressed, but the application of the method in this study is not explained clearly. For example, how the samples were collected? 3. Page 20, Fig. 7. Based on Figs. 6 and 7, the temporal change of NDVI mainly appears in

the climate transition from desert to steppe. The authors should address this in greater detail.

---

## Editor Comment (EC1) · DD Xing (Editor) · 3 Nov 2016

Agree with this reviewer and the comments given by this reviewer should be addressed in further revision.

---

## Editor Comment (EC2) · DD Xing (Editor) · 3 Nov 2016

I fully agree with this reviewer. The revision should reflect the comments given by this reviewer.

---

## Author Comment (AC1) · 14 Dec 2016

The comment was uploaded in the form of a supplement:
http://www.biogeosciences-discuss.net/bg-2016-376/bg-2016-376-AC1-supplement.zip

---

## Author Comment (AC2) · 14 Dec 2016

The comment was uploaded in the form of a supplement:
http://www.biogeosciences-discuss.net/bg-2016-376/bg-2016-376-AC2-
supplement.zip

---

## Author Comment (AC3) · 14 Dec 2016

The comment was uploaded in the form of a supplement:
http://www.biogeosciences-discuss.net/bg-2016-376/bg-2016-376-AC3-supplement.zip

---

## Author Comment (AC4) · 14 Dec 2016

The comment was uploaded in the form of a supplement:
http://www.biogeosciences-discuss.net/bg-2016-376/bg-2016-376-AC4-supplement.zip

---

## Author Comment (AC5) · 14 Dec 2016

The comment was uploaded in the form of a supplement:
http://www.biogeosciences-discuss.net/bg-2016-376/bg-2016-376-AC5-
supplement.zip